# Extracellular Vesicles Derived from *Opuntia ficus-indica* Fruit (OFI-EVs) Speed Up the Normal Wound Healing Processes by Modulating Cellular Responses

**DOI:** 10.3390/ijms25137103

**Published:** 2024-06-28

**Authors:** Anna Valentino, Raffaele Conte, Dalila Bousta, Hicham Bekkari, Anna Di Salle, Anna Calarco, Gianfranco Peluso

**Affiliations:** 1Research Institute on Terrestrial Ecosystems (IRET), CNR, Via Pietro Castellino 111, 80131 Naples, Italy; raffaele-conte@cnr.it (R.C.); anna.disalle@cnr.it (A.D.S.); gianfranco.peluso@unicamillus.org (G.P.); 2National Biodiversity Future Center (NBFC), 90133 Palermo, Italy; 3National Agency of Medicinal and Aromatic Plants Tounate, Taounate 34000, Morocco; boustadalila@gmail.com; 4Laboratory of Biotechnology, Environment, Agrofood and Health (LBEAS), Fez 30000, Morocco; hichambekkari@yahoo.fr; 5Faculty of Medicine and Surgery, Saint Camillus International University of Health Sciences, Via di Sant’Alessandro 8, 00131 Rome, Italy

**Keywords:** Opuntia, extracellular vesicles, phenols, flavonoids, wounds and injuries, cellular responses, antioxidant

## Abstract

Plant-derived extracellular vesicles (EVs) have been recognized as important mediators of intercellular communication able to transfer active biomolecules across the plant and animal kingdoms. EVs have demonstrated an impressive array of biological activities, displaying preventive and therapeutic potential in mitigating various pathological processes. Indeed, the simplicity of delivering exogenous and endogenous bioactive molecules to mammalian cells with their low cytotoxicity makes EVs suitable agents for new therapeutic strategies for a variety of pathologies. In this study, EVs were isolated from *Opuntia ficus-indica* fruit (OFI-EVs) and characterized by particle size distribution, concentration, and bioactive molecule composition. OFI-EVs had no obvious toxicity and demonstrated a protective role in the inflammatory process and oxidative stress in vitro model of chronic skin wounds. The results demonstrated that pretreatment with OFI-EVs decreased the activity and gene expression of pro-inflammatory cytokines (IL-6, IL-8, and TNF-α) in the LPS-stimulated human leukemia monocytic cell line (THP-1). Furthermore, OFI-EVs promote the migration of human dermal fibroblasts (HDFs), speeding up the normal wound healing processes. This study sheds light, for the first time, on the role of OFI-EVs in modulating important biological processes such as inflammation and oxidation, thereby identifying EVs as potential candidates for healing chronic cutaneous wounds.

## 1. Introduction

The largest organ in the body is the skin, which is composed of various types of cells that interact in a highly coordinated way to maintain homeostasis [1]. Cutaneous injuries, particularly chronic wounds, require long-term treatment and place an enormous economic burden on healthcare systems due to their prevalence in aging populations. Indeed, patients often presented underlying comorbidities, such as diabetes mellitus, peripheral vascular disease, connective tissue disease, or other systemic illnesses that contribute to impaired healing [2,3].

The healing of cutaneous wounds is a complex, multifactorial process involving numerous cellular and molecular effectors. It involves the orchestration in space and time of several highly controlled components working together to return damaged skin to its repaired barrier function. The great majority of superficial wounds follow four separate stages, which happen in an overlapping manner (i.e., hemostasis, inflammatory, proliferative, and remodeling phases). Chronic non-healing wounds can develop when the healing process does not proceed as expected, stagnating in the inflammatory reaction phase [3]. Indeed, prolonged inflammation causes imbalances in cytokine release that normally only occur for a short period. This imbalance can increase the degradation of growth factors and extracellular membrane (ECM) receptors [4]. Degradation products require a continuous immune response, initiating a cycle of persistent, non-healing inflammation [5]. The excessive workload of immune cells compromises their function, allowing bacteria to infiltrate the wound and form a multicellular biofilm. Research is focused on interrupting the cycle of inflammation and ECM breakdown, as well as removing the biofilm layer, to promote the healing of chronic wounds [6].

Over recent years, plant-derived extracellular vesicles (P-EVs) have emerged as a notably promising therapeutic approach for clinical use due to their wide availability, cost-effectiveness, and easiness to obtain. Moreover, P-EVs carry unique cargo loads (i.e., bioactive molecules, miRNA, proteins, and lipids), which in turn determine their biochemical functionality and play a role in interspecies communication [7,8]. Notably, P-EVs offer distinct advantages over existing delivery systems. P-EVs did not exert cellular toxicity, reducing adverse reactions, particularly when compared to synthetic liposomes and EVs sourced from animals [9]. Their small size and negative charge, along with heightened physicochemical stability across varying pH and temperature conditions, facilitate efficient penetration to target cells. Indeed, P-EVs have the capacity to penetrate cell membranes, release their cargo, and influence the response of recipient cells [10]. Their therapeutic effects have been proven in several human diseases, such as wound healing and tissue repair, relying on their powerful antioxidant and anti-inflammatory effects [11]. For instance, Yağız Savcı et al. demonstrated that grapefruit-derived extracellular vesicles (GEVs) reduce intracellular ROS levels under H_2_O_2_-induced oxidative stress in the human epidermal keratinocytes. This reduction promoted cell proliferation and migration, increasing the expression of mRNAs and proteins related to wound healing. Moreover, a tube formation assay confirmed that GEVs promoted capillary tube formation, thereby improving wound healing [12]. Similarly, Sánchez-López et al. [13] extracted EVs from pomegranate juice, underlining their anti-inflammatory effects on human leukemia monocytic cell lines, and their ability to enhance the healing process using a scratch assay. Additionally, EVs from *Aloe Saponaria* were applied to different cell lines to investigate chronic skin wound healing. Reported results showed no toxic effects; meanwhile, the mRNA expression of inflammatory cytokines, such as IL-6 and IL-1β, induced by lipopolysaccharide (LPS) stimulation, was suppressed. These EV-mediated anti-inflammatory effects were correlated with the increased proliferation and migration of human dermal fibroblast cells (HDFs) [14] in the presence of EVs. Wheat-derived EVs possess regenerative properties, enhancing the proliferation and migration of different cell types involved in the chronic skin wound healing process such as endothelial cells, epithelial cells, and dermal fibroblasts. This effect was associated with the increased expression of collagen type I mRNA, the formation of tube-like structures, and an apoptosis reduction [15]. 

*Opuntia ficus-indica* (OFI) pear fruit has attracted attention because it is rich in bioactive antioxidant compounds such as betalains, ascorbic acid, and polyphenols, which are known for their wellbeing properties. The nutraceutical benefits of the prickly pear fruit are believed to be antiulcerogenic, antioxidant, anticancer, neuroprotective, hepatoprotective, and antiproliferative [16,17,18]. In particular, studies have demonstrated that OFI fruit extract reportedly protects erythrocytes against lipid oxidation induced in vitro by ethanol [19]. Giraldo-Silva et al. demonstrated that scavenging activity and glutathione (GSH) levels were restored to near-normal levels after feeding rats with OFI prickly pear juice [20]. The normalization of scavenging activity via prickly pear juice supplements may be attributed to their natural antioxidant contents, which restore the equilibrium between oxidant species and the antioxidant defense system [21].

The present study aims to investigate, for the first time, the antioxidant and anti-inflammatory effects of OFI-derived extracellular vesicles (OFI-EVs) on chronic wound healing using in vitro experimental models. It is hypothesized that these beneficial effects may be due to the direct delivery of bioactive molecules stored within the EVs. 

The results obtained suggest that the use of OFI-EVs in the management of chronic wound healing represents a valuable future approach for regenerative purposes. Furthermore, due to their small size, low toxicity, high absorption, and environmental safety, plant-derived EVs could provide next-generation therapeutic delivery systems for the treatment of diseases other than skin diseases. However, given the site specificity and short plasma half-life of EVs, future research efforts should focus on EV formulation, exploring the synergistic potential of combining different biomaterials (i.e., hydrogel, electrospun membrane) with EVs to more comprehensively address the complex disease microenvironment.

## 2. Results and Discussion

### 2.1. OFI-EV Isolation and Characterization

Ultracentrifugation is a technique commonly used for isolating EVs from various macro- and micro-vesicles due to its ability to handle large-scale samples at low cost [22]. Ultracentrifugation can yield highly purified preparations by effectively separating EVs from other contaminants such as proteins, lipoproteins, and cellular debris, preserving the EVs’ structural and functional integrity, and ensuring that they remain intact and biologically active for downstream analyses. 

OFI-EVs were isolated from the entire fruit through the ultracentrifugation method as schematically depicted in Figure 1A. The physical and biomolecular characterization of OFI-EVs was performed according to the standards presented in MISEV2019 [23]. Based on the Nanoparticle Tracking Analysis (NTA) results, OFI-EVs exhibited a major peak range around 100 to 120 nm with a mean diameter of 114 ± 1.8 nm (Figure 1B). Moreover, NTA identified the presence of particles with a hydrodynamic diameter higher than 230 nm (approx. 4%), probably generated via EV aggregate formations. Dynamic light scattering (DLS) was used as a complementary approach to confirm size measurements (Figure 1C) and determine the zeta potential and polydispersity index (PDI). As shown in Figure 1D, OFI-EVs exhibited a negative surface charge of −23.8 ± 1.6 mV. Several studies have reported the presence of negative surface charge in plant and animal EV preparations [24,25]. The negative zeta potential allows OFI-EVs to interact with positively charged molecules or surfaces, such as mammalian cell membranes, facilitating cellular internalization. Moreover, the negative charge on EVs can promote selective adhesion to inflammatory regions, similar to the behavior observed with negatively charged liposomes. The PDI value of the EVs of 0.25 indicates a broad molecular weight distribution of the samples, consistent with the distribution plot. Next, the OFI-EV yield production was determined by dividing the number of isolated OFI-EVs by the weight of OFI used. As shown in Figure 1E, the production yield of OFI-EVs was 1.87 × 10^10^ particles per gram of OFI. The purity of OFI-EVs, obtained by dividing the number of OFI-EVs by total protein micrograms measured using a BCA assay, was 2.53 × 10^9^, as shown in Figure 1F. Finally, a transmission electron microscopy (TEM) analysis revealed that OFI-EVs were nearly spherical with an intact membrane. Moreover, TEM images confirmed that the EV size is around 100 nm in accordance with NTA results (Figure 1G). 

### 2.2. Phytochemical Analysis of OFI-EVs via HPLC-DAD-ESI/MS

Emerging evidence supports the fruit’s healthful properties in preventing the development of chronic diseases (i.e., diabetes, obesity, and respiratory diseases) due to the presence of several phytochemicals such as flavonoids, cinnamic derivatives, betalains, and tannins. In particular, *Opuntia ficus-indica* fresh fruit plays an important role in healthy diets, contributing to individuals’ daily intake of fibers, vitamins, and minerals [26]. Currently, research is being carried out on the analysis of phytoconstituents present in the whole OFI fruit or fruit peel. In a recent study, Bellumori et al. found an average of 0.45 mg/g of whole OFI fruit total phenolic content (TPC) with slight variations based on the harvesting period. The HPLC separation showed that flavonoids were the most abundant compounds, representing about 79% of the TPC, while phenolic acids were about 21% [27]. Differently from the total phenolic content, the polyphenol ratio was not influenced by the harvesting period or by the collection area. Similarly, Slimen et al. identified the polyphenolic profile of OFI fruit extract through LC/MS, with a TPC of 0.48 mg/g. Naringin, syringic acid, and trans-cinnamic acid were the most concentrated polyphenols while protocatechuic acid, catechin, coumaric acid, rutin, apigenin, and quercetin were detected in small amounts [28]. 

To the authors’ knowledge, this is the first study investigating the composition and biological activities of EVs isolated from whole OFI fruit. The total polyphenolic content of OFI-EVs was 0.066 mg/g of dry OFI extract. A spectrophotometric determination with Al(NO_3_)_3_ resulted in 0.04 mg/g of flavonoids; meanwhile, in using the Folin–Ciocalteu reagent, 0.02 mg/g of phenolic acids were quantified. Such values were confirmed via LC/MS, where a higher concentration of flavonoids (60.5%) was detected compared to phenolic acids (39.5%). 

Figure 2 shows the amount (mg/g) of phytocompounds present in OFI-EVs with respect to hydroalcoholic fruit extracts (%). Phenolic compound identification was performed by comparing them with the analytical standards when available and retention times and by comparing their mass spectral characteristics with that reported in the literature. In particular, kaempferol-3-O-pentose (89.4%), chlorogenic acid (78.4%), arbutin (78.9%), syringic acid (72.5%), and apigenin (69.1%) were the main compounds. Other phenolic compounds were detected in low concentrations, including quercetin-3-O-glucoside (26.5%), sinapic acid (24.8%), ferulic acid (13.9%), trans-ferulic acid (12.2%), rutin (10.3%), kaempferol-3-O-glucose (9.4%), quercetin-3-O-hexose deoxyhexose (9.6%), isorhamnetin- 3-O-rutinoside (2.9%), and caffeic acid (2.6%). 

Among the reported flavonoids, the isorhamnetin moiety flavonol is one of the most abundant, present in at least five different di- and triglycosides [29]. These flavonols showed anti-inflammatory and chemopreventive effects [29]. However, the OFI flavonol profile has been examined more in cladodes than in fruits, where few studies have been conducted [30,31]. In prickly pear juices as regards phenolic acids, hydroxybenzoic, piscidic, and caffeic acids and ferulic and piscidic acid derivatives have been identified [31]. Previous studies have demonstrated the efficacy of flavonoids in shortening the wound healing time by influencing collagen breakdown and MMP-2 activity following 24 h therapy. In particular, it was suggested that kaempferol presented an effective wound healing agent because of its ability to induce a healing rate of about 90% [32]. In another study, quercetin-loaded liposomal hydrogel was able to accelerate fibroblast cell growth in only 4 days of treatment on subjects with a shaved back wound. Additionally, one study demonstrated a reduction in wound area between the 4th and 16th days following treatment with a rutin-loaded hydrogel [32,33]. 

### 2.3. OFI-EVs Exhibit In Vitro Biological Activities

#### 2.3.1. Biocompatibility and Internalization by Living Cells

Plant-derived extravesicles have demonstrated excellent safety and biocompatibility in both in vitro and in vivo applications [34]. However, whenever new compounds are destined for human consumption, carrying out an in vitro cytotoxicity test is convenient. 

To assess in vitro cytotoxicity, OFI-EVs were incubated at different concentrations, with three cell types involved in the wound healing process: human dermal fibroblasts (HDFs), which contribute to the reconstitution of damaged dermal tissue; human umbilical vein endothelial cells (HUVECs), which are involved in angiogenesis; and the human monocytic cell line THP-1. 

The results, depicted in Figure 3A,C,E demonstrated that OFI-EVs did not exert any significant cytotoxic effect on HFDs, HUVECs, or THP-1 cells even after 72 h of incubation at the tested concentrations (5 and 20 µg/mL). In addition, the low LDH level in the cell supernatant confirms the absence of cell membrane damage (Figure 3B,D,F).

These data are consistent with several studies supporting the idea that EVs derived from edible plants are nontoxic. For example, increasing concentrations of pomegranate EVs (2.5, 5, 10, and 20 μg mL^−1^) did not significantly alter the cell viability of Caco-2 and THP-1 cells after 48 h [35]. No detectable cytotoxicity was observed when human and mouse cells (HaCaT, HDF, and RAW264.7 cells) were incubated in the presence of cabbage and red cabbage EVs. 

To exert their biological function, plant-derived EVs need to interact with and be internalized by mammalian cells. EVs can fuse directly with the plasma membrane of recipient cells, releasing their cargo into the cytoplasm. This uptake mechanism is mediated by specific interactions between proteins and lipids on the EV surface and the target cell membrane [36]. The specific mechanisms and pathways involved in EV uptake may vary depending on factors such as the cell type, the composition of the EVs, and the physiological or pathological context. 

To confirm the internalization of EVs by HDF cells, OFI-EVs (20 µg/mL) were fluorescently labeled with the lipophilic green dye PKH67. Briefly after incubation with EVs, a punctuated pattern of fluorescence could be observed in cell cytoplasm, with a fluorescence signal accumulation with increasing incubation time (Figure 3G). A semi-quantitative analysis of fluorescence intensity in the HDF cytoplasm is reported in Figure 3H.

#### 2.3.2. OFI-EVs Show a Cytoprotective Effect against Oxidative Stress Induced by Hydrogen Peroxide

Several studies have highlighted a positive correlation between a diet rich in plant-based foods and a reduced risk of oxidative stress-related diseases. Antioxidants help to neutralize harmful molecules called free radicals, which can damage cells and contribute to aging and disease. OFI fruits contain phytochemical compounds, particularly kaempferol, arbutin, apigenin, and a flavonoid fraction known to have antioxidant properties [9,37].

The efficiency of OFI-EVs in reducing intracellular ROS production was assessed in HDF cells in the presence of hydrogen peroxide (H_2_O_2_) (Figure 4A,B). A significant increase (*p* < 0.005) in fibroblast intracellular oxidants of about 2.4 times was obtained with respect to untreated cells (control) after H_2_O_2_ 24 h treatment (Figure 4A). A short-time pre-incubation (24 h) with OFI-EVs considerably reduced (*p* < 0.01) H_2_O_2_-induced ROS production by approximately 1.4-fold with respect to the H_2_O_2_ group. Moreover, the protective effect of OFI-EVs was greatly enhanced (*p* < 0.005) with a higher concentration of 20 µg/mL resulting in a slight fluorescence increase compared to control cells. Under oxidative stress conditions, the main targets of ROS are polyunsaturated fatty acids of cell membrane lipids. Lipid peroxidation leads to the formation of chemically reactive species, such as MDA, able to damage nucleic acids and proteins, causing the loss of both structural and metabolic functions of cells [38]. As reported in Figure 4B, the treatment of cells with H_2_O_2_ increased intracellular lipid peroxidation to 2-fold relative to control (*p* < 0.005). Conversely, the presence of OFI-EVs for 24 h markedly diminished (*p* < 0.01) the MDA level (1.1-fold) with respect to H_2_O_2_-treated cells, with a marked decrease (*p* < 0.005) in the presence of 20 µg/mL leading the MDA formation to reach levels almost like those of the control.

In order to mitigate the accumulation of damage caused by reactive oxygen species (ROS), cells generate a range of antioxidant enzymes, such as superoxide dismutases (SODs), catalase (CAT), and glutathione (GSH) [39]. The three SOD family members—SOD1, SOD2, and SOD3—transform O^2-^ into hydrogen peroxide (H_2_O_2_), limiting the formation of highly aggressive compounds such as ONOO^−^ and OH^−^. All SODs are expressed at lower levels in stress conditions with respect to the normal state, at both mRNA and protein levels. To verify whether the OFI-EVs antioxidant effects have been related not only to its free radical scavenging activity but also to the ability to improve the endogenous defense system by influencing antioxidant/detoxifying enzyme activity, SOD2 activity was tested. As demonstrated in Figure 4C, H_2_O_2_ treatment leads to a decrease in antioxidant enzyme activity of about 44% compared to untreated cells. When fibroblasts were pretreated with OFI-EVs (5 µg/mL and 20 µg/mL), SOD2 activity increased by 38% and 52%, respectively; thus, H_2_O_2_-treated cells show a fine ability to protect mitochondria against oxidative damage. Furthermore, OFI-EV pretreatment restored the SOD2 transcription to above their control levels by significantly increasing its expression by 1.4-fold with 5 µg/mL and 3.5-fold with 20 µg/mL over the H_2_O_2_-depressed level (Figure 4D). Overall, the data reported herein confirm the crucial role of OFI-EV pretreatment to suppress the production of intracellular ROS and lipid peroxidation as well as the capability to elevate the activity of antioxidant enzymes such as SODs, limiting the damage induced by oxidative stress in the HDF model.

#### 2.3.3. OFI-EVs Anti-Inflammatory Property

During a normal wound healing process, inflammation typically serves to remove initial triggers and kickstart the regeneration of damaged tissues through a coordinated immune response, notably involving macrophages and mast cells [40,41]. However, under certain circumstances, the mechanisms responsible for restoring tissue equilibrium falter, leading to an uncontrolled reaction that impairs the transition from inflammation to re-epithelialization [42,43]. In addition, protracted inflammation can raise the ROS levels in the wound lesion, leading to the activation of pro-inflammatory genes. According to several studies, the use of naturally derived antioxidants promotes the healing of chronic wounds, including skin wounds. Therefore, the potential effect of OFI-EV in chronic wound healing was investigated in THP-1 cells, which are implicated in innate immunity responses. To this end, cells were treated with OFI-EVs at concentrations of 5 and 20 µg/mL before the induction of an inflammatory response with LPS. 

As expected, the pretreatment of cells for 24 h with OFI-EVs significantly decreased the number of released cytokines (interleukin (IL)-6, interleukin (IL)-8, and TNF-α) compared to untreated cells (Figure 5A,C,E). Consistently, mRNA expression levels of pro-inflammatory cytokines confirmed the protective effects of OFI-EVs in treated cells with respect to non-treated ones. As shown in Figure 5B,D,F, LPS stimulation induced a 2-fold increase for IL-6 and IL-8, and 2.5 for TNF-α in the mRNA expression levels. However, pretreatment with OFI-EVs led to more than a 45% reduction in gene expression, limiting the inflammatory response induced by LPS treatment. 

Several studies report the anti-inflammatory effect related to OFI extracts. Smeriglio et al. described the ability of OFI extract power to reduce intestinal inflammation compared to dexamethasone and trolox [44,45]. The Opuntia anti-inflammatory ability has been demonstrated in a Moroccan study. The anti-inflammatory activity of extracted seed oil from OFI was evaluated in Swiss rats subjected to chemical injury. The study showed that the use of this seed oil demonstrated anti-inflammatory effects on inflammation induced by trauma, specifically reducing paw edema in female rats [44,45,46]. Again, both betalains and OFI betalain-pure extract decreased the release of important inflammatory markers such as IL-6, IL-8, and NO, efficiently reducing the intestinal inflammation state [44]. Another study reported the Opuntia flower’s anti-inflammatory potential; the authors found that OFI flowers’ methanolic extracts showed anti-inflammatory potential by reducing the size of paw edema in Wistar rats, as effectively as the nonsteroidal anti-inflammatory drug indomethacin [47]. Imene Ammar and colleagues [48] conducted a study showing that polyphenols extracted from various parts of OFI fruits, including seeds, pulp, and the whole fruit, exhibited anti-inflammatory properties. These polyphenols were found to potentially offer neuroprotective effects by reducing the transcriptional expression of pro-inflammatory mediators like TNF-α, IL-1, and iNOS. This study was conducted on N13 microglial cells after stimulation with lipopolysaccharides, highlighting the potential health benefits of these compounds following neuronal damage. In a study utilizing the intestinal Caco-2/TC7 cell line, Filannino and colleagues [49] found that extracts derived from raw and fermented OFI cladodes demonstrated anti-inflammatory properties. These extracts notably reduced the production of nitric oxide (NO) as well as key chemokines such as IL-8 and TNF-α, which play crucial roles in the inflammatory process by recruiting and activating various inflammatory cells.

#### 2.3.4. OFI-EVs Accelerate/Facilitate Wound Healing Process and Induce Angiogenesis

When the inflammatory response in the skin has subsided, the proliferation and migration of cells such as fibroblasts and keratinocytes are essential steps in the process of healing cutaneous wounds [50]. Unlike regular wounds, chronic skin wounds are distinguished by an increase in the activity of matrix metalloproteinases (MPPs), which degrade proteins of the extracellular matrix (ECM) [51]. The excessive activation of MPPs impedes the migration of fibroblasts and hinders the healing process by creating a proteolytic environment that is detrimental to wound recovery [52]. To assess the impact of OFI-EVs on fibroblast migration after serum starvation to minimize the stimulatory effect induced by serum, HDFs were treated with 20 µg/mL of OFI-EVs. A wound healing assay showed that under normal conditions, the untreated cells migrate to healing the wound after 24 h. In contrast, the presence of stress conditions induced no cell migration toward the center with a significant reduction in wound closure. HDF OFI-EV treatment closed the gap faster than untreated cells (Figure 6A). The quantitative analysis of the wound assay demonstrated the stimulatory effect of OFI-EVs on HDF migration, with a wound recovery of about 85% compared to 38% of H_2_O_2_ (Figure 6B).

Skin wound repair is involved in several biological interactions, aimed to regulate cellular behaviors and to reorganize the extracellular matrices in order to improve the regeneration and repair processes [42]. Typical mechanisms that modulate cell behaviors to promote cutaneous wound healing are matrix remodeling, re-epithelialization, proliferation and migration, hemostasis, and angiogenesis [53]. Angiogenesis is a natural process of the proliferation phase involved in acute wound healing but could be challenging in a chronically inflamed wound [54].

The tube formation assay is a widely used in vitro technique to assess angiogenesis, the process by which new blood vessels are formed. This assay is particularly useful for studying the ability of endothelial cells to organize into tubular structures, resembling capillaries, which is a critical step in the formation of functional blood vessels. Several natural compounds present pro-angiogenic activity, promoting faster vessel growth and quick recovery in healing tissue [6]. These molecules exert their effects through various mechanisms, including stimulating endothelial cell proliferation, migration, and tube formation. Natural bioactive molecules offer potential therapeutic benefits for conditions characterized by impaired angiogenesis, such as ischemic heart disease, peripheral artery disease, and chronic wounds. However, further research is needed to fully understand their mechanisms of action and assess their efficacy and safety in clinical settings.

A number of pathological conditions are known to be associated with aberrant angiogenesis, such as chronic skin wounds. Tube formation is maintained for 18–24 h, after which apoptosis is activated, leading to the disintegration of the tube networks. As shown in Figure 6C when HUVECs were treated with OFI-EVs (20 µg/mL), the number of branches increased significantly (*p* < 0.005) with respect to positive CTL with vascular endothelial growth factor (VEGF). The quantitative analysis of HUVEC cells’ number of branches for fields treated with OFI-EVs was increased significantly by about 50% with respect to the control as shown in Figure 6D.

## 3. Materials and Methods

### 3.1. Opuntia ficus-indica Extravesicle Isolation

Prickly pears were collected in September 2022 from farms in Calabria. Samples were transferred to the laboratory in refrigerated boxes at 4 °C and processed upon arrival. Before peeling, the fruits were thoroughly washed to remove impurities and spines.

OFI fruits were spliced into small pieces and then chopped in a regular blender. A volume twice the weight of vesicle isolation buffer (VIB: MES 20 mM, NaCl 100 mM, and CaCl_2_ 2 mM) was added to the chopped fruits and left under stirring overnight at room temperature. Then, the mixture was subjected to differential ultracentrifugation. A typical UC-based workflow for EV isolation is the following: centrifugation at 400× *g* for 10 min to sediment the main portion of fruit cells; centrifugation of the supernatant at 3000× *g* to remove cell debris and discard the pellet. The resulted supernatants were centrifuged at 10,000× *g* to remove the aggregates such as apoptotic bodies and other structures with a buoyant density higher than that of EVs; also in this case, the pellet is discarded. Finally, there is the ultracentrifugation of the supernatant at 150,000× *g* for 2 h to pellet EVs using a 70Ti rotor. The pellet of EVs is resuspended in phosphate-buffered saline (PBS) and stored at −80 °C for further analysis.

### 3.2. OFI-EVs Characterization

All the experimental procedures for the evaluation of extravesicle physicochemical properties and biocompatibility, such as transmission electron microscopy (TEM), nanoparticles tracking analysis (NTA), and dynamic light scattering (DLS), are reported in the Supporting Information. 

### 3.3. (Poly)phenolic Content of OFI

The Total Phenol Content (TPC) and Total Flavonoid Content (TFC) methods are described in detail in the Appendix A [55]. For unambiguous polyphenol chromatographic analysis and separation, 80 mg of dried OFI-EVs extract was resuspended in 10 mL of ethanol. The mixture was sonicated for 60 min at 45 °C. For efficient injection into the bulk system, further dilution of the sample with acetonitrile (1:20 *v*/*v*) was required. A Shimadzu Ultra-High-Performance Liquid Chromatograph (Nexera XR) combined with an MS/MS detector (LCMS 8060, Shimadzu Italy, Milan, Italy) was used to determine the phenolic profile of the sample. Electrospray ionization in negative mode was employed for detection. In particular, chromatographic separation was obtained in isocratic conditions using acetonitrile/water + 0.01% formic acid (5:95, *v*/*v*) as mobile phase and Kinetex 2.6 µm C18 100 Å, LC Column 100 × 4.6 mm (Phenomenex Inc., Torrance, CA, USA) as stationary phase. Mass conditions were set as nebulizing gas flow: 3 L/min; heating gas flow: 10 L/min; interface temperature: 300 °C; DL temperature: 250 °C; heat block temperature: 400 °C; and drying gas flow: 10 L/min.

### 3.4. Biological Activity

#### 3.4.1. In Vitro Cell Model

Human dermal fibroblasts (HDFs), primary human umbilical vein endothelial cells (HUCECs), and human leukemia monocytic cell line (THP-1) were provided from ThermoFisher Scientific (Roma, Italy) and maintained in complete medium (Dulbecco’s Modified Eagle Medium (DMEM)), Endothelial Cell Growth Medium, and RPMI-1640-based medium, respectively. HDF- and THP-1-cultured media contain 10% of fetal bovine serum, and antibiotics such as penicillin (100 U/mL) and streptomycin (100 mg/mL); meanwhile, HUVECs were grown in low-serum (2%) medium without phenol red. Both cell lines grew at 37 °C in a humidified atmosphere with 5% CO_2_. The cells were checked periodically by performing contamination tests, including the presence of Mycoplasma. When the confluence was about 80%, the experiments were carried out. The protective effects of OFI were studied with an acute toxicity model of chronic skin wound by pretreating cells with OFI-EVs (5 µg/mL and 20 µg/mL) for 24 h followed by a 24 h H_2_O_2_ (230 µM) [56] treatment for the antioxidant assay and LPS (1 μg/mL) for the anti-inflammatory assay [57]. mRNA expression was evaluated after a shorter OFI-EV exposure (4 h).

#### 3.4.2. Cell Proliferation and Migration Assay

According to the manufacturer’s protocols, the solution (10 µL/well) of CCK-8 was added to human dermal fibroblast cells seeded in a 96-well plate. Following the reaction with water-soluble tetrazolium salt, the reaction product formed water-soluble formazan dye and was read at 450 nm in a microplate reader, Citation 3 Cell Imaging Multi-Mode (ASHI, Milan, Italy). In particular, cells prior to conducting the experiment were incubated overnight in low-serum media (0.1% FCS). Then, dermal fibroblasts were exposed to media containing OFI-EVs at 5 µg/mL and 20 µg/mL for 24, 48, and 72 h. The experiment was conducted in triplicate. Lactate dehydrogenase (LDH) release measurements were based on the measurement of lactate LDH released into the growth media when the integrity of the cell membrane was lost. For this assay, HDF, HUVEC, and THP-1 cells were treated with OFI-EVs in the same way as described above. At the end of the incubation time, 100 μL of the culture supernatants was collected, and LDH activity was detected as described by Di Salle et al. [58].

For the scratch migration assay, HDF cells were seeded on a 96-well plate (8 × 10^3^/well) and cultured to 90% confluence using the above-described protocol. Then, a sterile pipette tip (200 µL) was used to inflict a scratch wound (T0). Following the PBS wash, cells were incubated with OFI-EV (20 µg/mL)-containing medium for 24 h (T24). A proliferation inhibitor 0.1% mM mitomycin c, was also added (Merck Millipore, Milan, Italy). To photograph dermal fibroblast migration was used an inverted phase-contrast microscope (Zeiss, Milan, Italy) and then, the percentage of wound closure was calculated according to the following equation:$$Wound\;closure\;\left(\%\right)=\frac{A0-At}{A0}\times 100$$
where A0 represents the wound area recorded at h 0 and ImageJ software Version 1.54j was used to measure the wound area recorded after 24 h. The experiment was conducted in triplicate.

#### 3.4.3. OFI-EV Cellular Uptake

The labeling of EVs with green PKH67 was performed by applying the manufacturer’s instructions. Briefly, the final 2X dye solution was prepared by adding 4 μL of the ethanolic PKH67 dye solution to 1 mL of diluent C. The resulting solution was then transferred to a centrifuge tube and mixed well to homogenize the sample. An EV/dye solution (1:1 dilution EVs/dye, no less than 100 μg of EVs suspended in 100 μL of PBS) was incubated at room temperature for 10 min and then ultracentrifuged at 120,000× *g* for 60 min at 4 °C to avoid free dye in the suspension and to eliminate the excess of dye in the EVs membrane. The staining of the EVs was performed a maximum of 3 h prior to the treatment. This experiment was performed with OFI-EVs at 20 µg/mL. 

#### 3.4.4. Intracellular Oxidative Stress

DCFH-DA assay was used to measure the production of intracellular reactive oxygen species (ROS) in HDF cells with or without OFI-EVs (5 and 20 µg/mL) according to the manufacturer’s protocol. Following treatment, cells were labeled with DCFH-DA (25 μM) for 1 h in the dark. The fluorescence was measured every 5 min for 1 h, with an excitation wavelength of 485 nm and an emission wavelength of 535 nm using a microplate reader (Cytation 3 Cell Imaging Multi-Mode (ASHI, Milan, Italy)). 

The thiobarbituric acid-reactive substance (TBARS) assay was used to determine malondialdehyde (MDA) concentration, as a lipid peroxidation index. The basal concentration of MDA was established by adding about 600 μL of TBARS solution to 50 μg of total protein dissolved in 300 μL of Milli-Q water. Before centrifuging (14,000 rpm for 2 min), the mix was incubated for 40 min at 100 °C. The supernatant was analyzed with a microplate reader at a wavelength of 532 nm. 

Total SOD2-like activity was assessed with the SOD2 Assay Kit-WST according to the manufacturer’s protocol. The activity was expressed as units per mg of protein, where one unit of enzyme inhibited the reduction in cytochrome C by 50% in a coupled system formed by xanthine and xanthine oxidase.

#### 3.4.5. Enzyme-Linked Immunoabsorbent Assay (ELISA)

Secreted IL-6, IL-8, and TNF-α protein levels were measured in supernatants of human dermal fibroblast treated with OFI-EVs (5 and 20 µg/mL) stimulated with LPS (1 µg/mL). To the wells already precoated with antibodies specific for IL-6, IL-8, or TNF-α, 100 µL of samples and standards was added and then incubated for 2 h at 37 °C following the procedure described by Valentino et al. [59]. 

#### 3.4.6. Real-Time Quantitative PCR (RT-qPCR)

Real-Time Quantitative PCR (RT-qPCR) evaluated anti-inflammatory activity according to the manufacturer’s protocols as described in the Supporting Information. Cells were treated with and without OFI-EVs (5 and 20 µg/mL) and stimulated with LPS (1 µg/mL) before RNA extraction. Primers used for qRT-PCR are listed in Appendix A.

#### 3.4.7. Tube Formation Assay

To investigate the tubule formation activity of HUVECs under conditioned media (OFI-EVs 20 µg/mL), the formation of capillary-like structures on a growth-factor-reduced MatrigelTM matrix (Corning) was assessed. HUVECs were incubated for 3 h in the factor-free experimental media containing 5% FBS prior seeded on MatrigelTM. Positive control media was obtained by adding 20 ng/mL of VEGF in the experimental media. Matrigel was thawed at 4 °C overnight, transferred into a 48-well plate (120 µL/well) with a cold tip, and allowed to polymerize at 37 °C for 30 min. During Matrigel incubation, HUVECs were tripsinized, resuspended in 5% FBS experimental media, counted, and a suspension of 3 × 10^6^ cells/mL was prepared. Following Matrigel polymerization, 280 µL/well of pre-warmed experimental media was added, as well as 20 µL/well of the cell suspension, to achieve a final concentration of 60,000 cells/well. After gently shaking the plate to uniformly distribute cells within the wells, plates were introduced in the CO_2_ incubator and imaged after 4 h and 8 h with the inverted phase-contrast microscope Nikon TE200 using the DP72 Olympus camera. From the triplicates used for each condition, three pictures at random sides were taken. Finally, an image analysis was performed with the “Angiogenesis Analyzer” plugin developed by Gilles Carpentier for the open-source software Image J, Version 1.54j.

### 3.5. Statistical Analysis

Results of biochemical and biological assays were calculated according to a one-way analysis of variance (ANOVA) with Tukey’s post hoc test. The statistically significant difference was accepted when *p* < 0.05. The data were analyzed using the statistical software package GraphPad Prism version 6.01 (GraphPad, San Diego, CA, USA). Triplicates were performed for each experiment. Results were reported as means ± standard deviation (SD).

## 4. Conclusions

Research conducted both in vitro and in vivo has shown that plant-derived extracellular vesicles (P-EVs) exhibit intrinsic therapeutic properties such as anti-inflammatory, antioxidant, and anti-tumor activities. Furthermore, P-EVs have no significant toxicity and cannot activate host immune responses. Due to the above-mentioned advantages, it is expected that P-EVs will have excellent and strong competition in clinical applications or preventive healthcare products in the future. This work demonstrated, for the first time, that extracellular vesicles isolated from *Opuntia ficus-indica* (OFI-EVs) fruits recover in vitro inflammatory and oxidative damage related to chronic skin wounds, thereby speeding up the wound healing process. These results suggest that OFI-EVs may have potential as a natural drug delivery system for the treatment of oxidative stress disorders and skin regeneration. However, given the site specificity and short plasma half-life of EVs, future research efforts should focus on EV formulation, exploring the synergistic potential of combining different biomaterials (i.e., hydrogel, electrospun membrane) with EVs for the realization of the full potential of these systems.

## Figures and Tables

**Figure 1 ijms-25-07103-f001:**
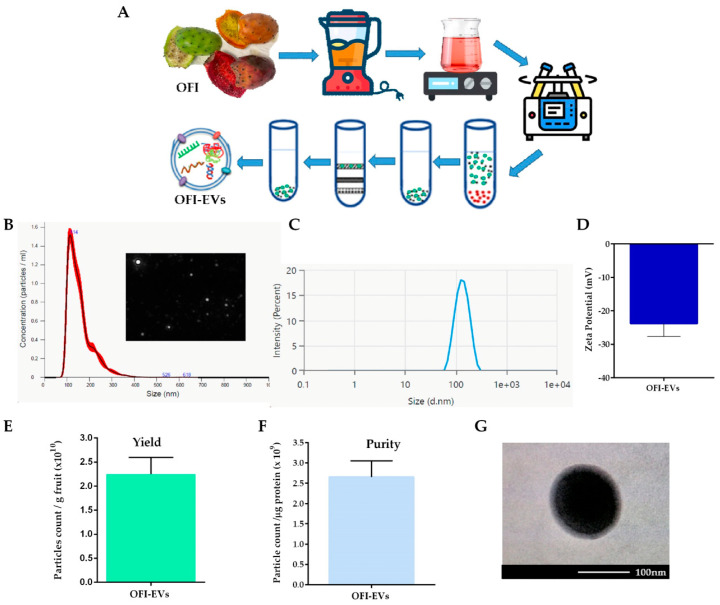
Isolation and characterization of *Opuntia ficus-indica*-derived extracellular vesicles (OFI-EVs). (**A**) Schematic representation of OFI-EVs isolation procedure. (**B**) Size distribution and representative screenshot video of OFI-EVs measured via Nanoparticle Tracking Analysis (NTA). (**C**,**D**) OFI-EV size distribution and zeta potential determined using dynamic light scattering (DLS). (**E**,**F**) OFI-EV yield production and purity. (**G**) OFI-EV transmission electron microscopy image.

**Figure 2 ijms-25-07103-f002:**
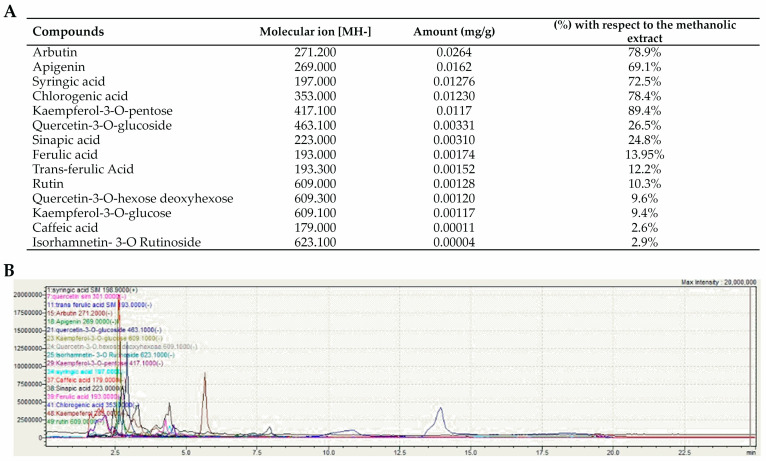
Phytochemical analysis of OFI-EVs. (**A**) Quantification (mg/g dry extract) and molecular ion [MH-] of phenolic compounds in OFI-EVs with respect to prickly pear (*Opuntia ficus-indica* L. Mill.) hydroalcholic extracts (%). Data are obtained in three independent experiments and represented as means ± SEM (standard error of the mean). (**B**) MS spectrum obtained by averaging the spectra over the chromatographic peak of OFI-EV bioactive molecules.

**Figure 3 ijms-25-07103-f003:**
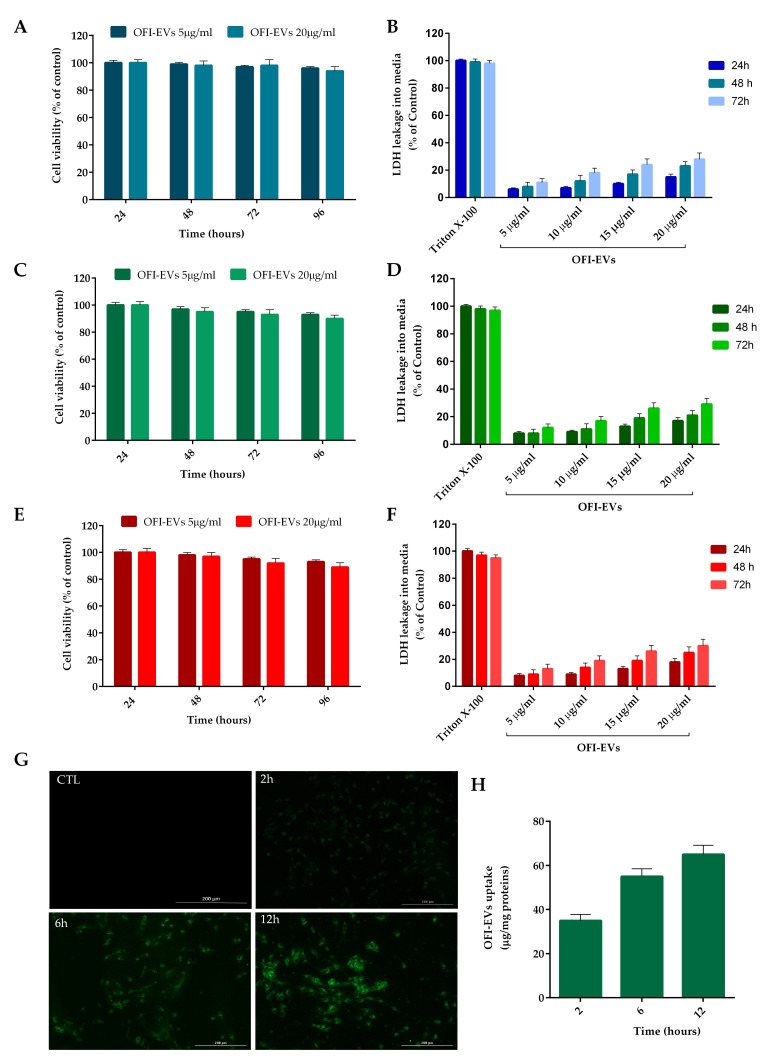
In vitro biocompatibility and uptake of OFI-EVs. (**A**) Cytotoxicity was determined in HDFs, (**C**) HUVECs, and (**E**) THP-1 cells after 24, 48, and 72 h of incubation with OFI-EVs 5 and 20 µg/mL. A Lactate dehydrogenase (LDH) assay was performed in HDFs (**B**), HUVECs (**D**), and THP-1 (**F**) cells with 5, 10, 15, and 20 µg/mL. Untreated cells were used as controls. (**G**) EV internalization at different time points. Cells were treated with 20 µg/mL of OFI-EVs dyed with PKH67 (green fluorescent dye). Negative control was without EVs. (**H**) Semi-quantitative analysis of PKH67-EVs fluorescence intensity in the cytoplasm was evaluated with respect to µg/mg proteins. Data are expressed as the means of three independent experiments ± S.D (*n* = 3).

**Figure 4 ijms-25-07103-f004:**
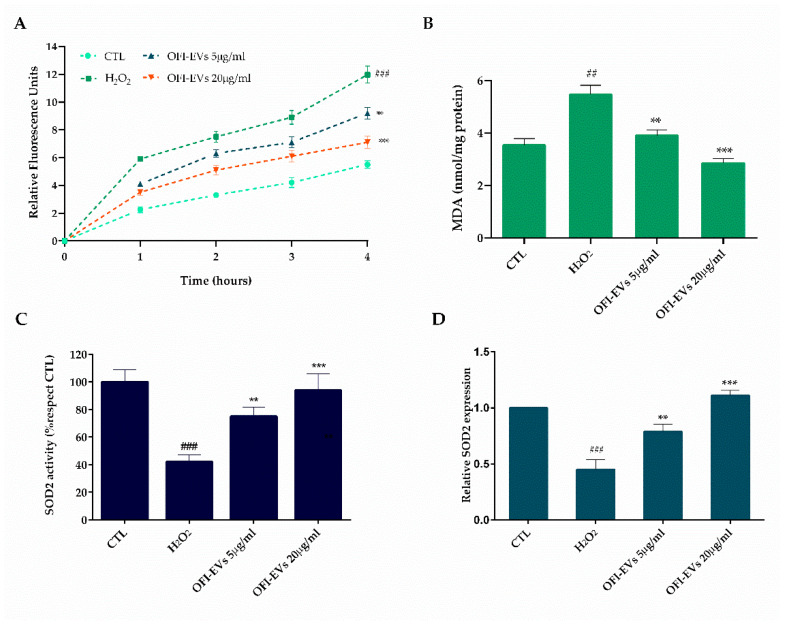
Antioxidant capacity of OFI-EVs in H_2_O_2_-treated HDF cells, in the presence of OFI-EVs for 24 h and then treated with H_2_O_2_ for 24 h. (**A**) Oxidized H2DCFDA (DCF) was used to determine ROS release. (**B**) MDA quantity was utilized as a marker of lipid peroxidation. (**C**) Superoxide dismutase (SOD2) activity. (**D**) SOD2 mRNA transcription level. Results are expressed as the means of three independent experiments ± S.D (*n* = 3). ** *p* < 0.01, *** *p* < 0.005 vs. H_2_O_2_; ## *p* < 0.01, ### *p* < 0.005 vs. CTL.

**Figure 5 ijms-25-07103-f005:**
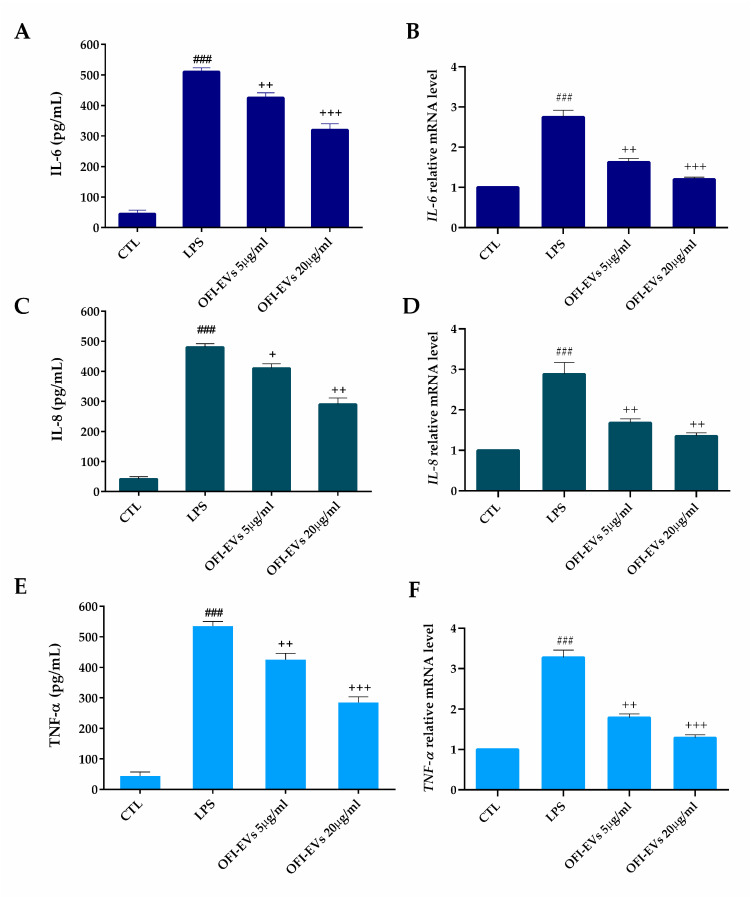
OFI-EVs inhibit LPS-induced inflammatory response in the THP-1 cell line. The effect of OFI-EVs (5 and 20 µg/mL) on the production of IL-6 (**A**,**B**), IL-8 (**C**,**D**), and TNF-α (**E**,**F**) was measured via an ELISA assay (**A**,**C**,**E**) and qRT-PCR (**B**,**D**,**F**). THP-1 cells were pretreated with OFI-EVs for 24 h, then stimulated with LPS (1 µg/mL) for 24 h (ELISA assay) or 4 h (qRT-PCR). Results are expressed as the means of three independent experiments ± S.D (*n* = 3). ### *p* < 0.005 vs. CTL, ^+^ *p* < 0.05, ^++^ *p* < 0.01, and ^+++^ *p* < 0.005 OFI-EVs vs. LPS.

**Figure 6 ijms-25-07103-f006:**
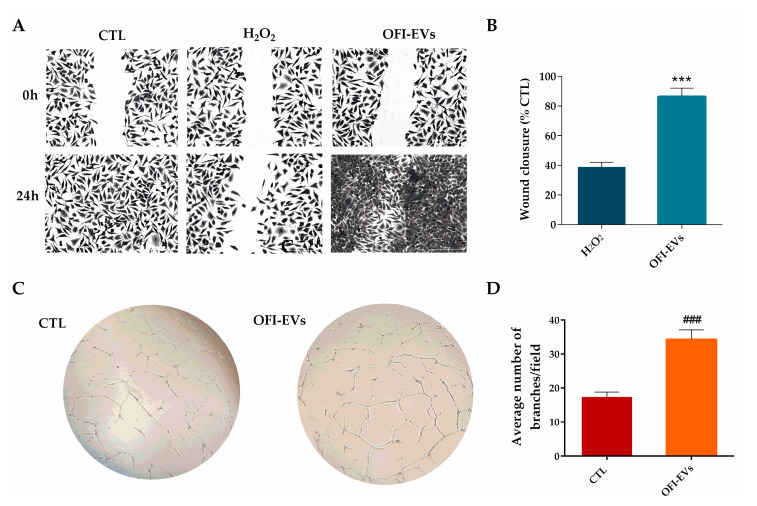
OFI-EVs speed up angiogenesis and wound healing process. (**A**,**B**). Wound closure representative images (0 and 24 h) after treatment with H_2_O_2_ alone or with OFI-EVs and relative quantitative analysis. Scale bars are 50 µm. (*n* = 3). (**C**) Optical images of HUVEC tubes at 4 h of incubation time of CTL (positive control treated with VEGF 20 ng/mL) and OFI-EVs; the image magnification was 20×. (**D**) Tube quantification with Image J software Version 1.54j Angiogenesis analyzer. *** *p* < 0.005 OFI-EVs vs. H_2_O_2_, ### *p* < 0.005 OFI-EVs vs. CTL. Data are expressed as means of three independent experiments ± S.D (*n* = 3).

## Data Availability

The data presented in this study are available in the article.

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
