# Peer review of "Extracellular Vesicles Derived from Opuntia ficus-indica Fruit (OFI-EVs) Speed Up the Normal Wound Healing Processes by Modulating Cellular Responses"

_ijms, 2024, doi:10.3390/ijms25137103_

Round 1

Reviewer 1 Report

Comments and Suggestions for Authors

I wish to thank the author's effort and their idea.

Some minor comments hope to be taken into consideration.

1. Fig 1G: Does it describe 100 nm or 100 μm? Kindly revise it; if you have a low magnification of the particles, I hope to insert a more obvious image.

2. Also, Fig 1. The resolution is inferior; the authors can insert another one in a good resolution without a shadow background.

3. in section 2.2. Kindly the symbol of the Oxy linkage of glycosides was written in italics. These mistakes are repeated more times at the end of page 5. And should be followed with a dash after that, not a dot.

4. Figure 2. B. The authors inserted the MS spectrum of the identified compounds and didn't refer to the molecular ion and the fragments of any compounds.

5. Figure 3, 5, and should be in good background resolution.

6. The authors fail to discuss the importance of the identified compounds to recover the wound healing activity.

7. Kindly, I hope the authors illustrate the Nano preparation part in a separate section in the methods part.

Author Response

Response to reviewers’ comments

We would like to thank the reviewer for the careful and thorough reading of this manuscript, and for the thoughtful comments and constructive suggestions, which helped to improve the quality of our work. Please see below, in red, our detailed response to comments. The requested changes have been reported in red in the manuscript.

Reviewer 2 Report

Comments and Suggestions for Authors

The manuscript is titled: “Extracellular vesicles derived from Opuntia ficus-indica fruit (OFI-EVs) speed up the normal wound healing processes by modulating cellular responses”. The researchers were able to extract the OFI-EVs by ultracentrifugation with a production yield of 1.87 × 1010 particles /g of OFI, purity 2.53 × 109 µg of total protein according to BCA assay, with spherical particles, of size range around 120 nm and sufficient surface charge -23 mV to impart stability against aggregation.

The phytochemical assay of OFI-EVs using LC/MS demonstrated that the chief ingredients present in the vesicles were kaempferol-3-O-pentose (89.4%), chlorogenic acid (78.4%), and arbutin (78.9%).

The biocompatibility of OFI-EVs was assessed through an in vitro cytotoxicity after 24, 48, and 72 h of incubation with three types of cells: human dermal fibroblasts (HDFs), human umbilical vein endothelial cells (HUVECs), and the human monocytic cell line THP-1 at two concentration levels (5 and 20 μg/mL).

The authors proved the active role of OFI-EVs in promoting the migration of human dermal fibroblast using scratch closure and transwell migration assays and increased the tube formation capacity of Human Umbilical Vein Endothelial Cells (HUVEC). In addition, pre-treatment of lipopolysaccharide LPS-stimulated human leukemia monocytic cell line (THP-1) with OFI-EVs decreased the activity and gene expression of proinflammatory cytokines (IL-6, IL-8 and TNF-α). Moreover, the high scavenging activity towards reactive oxidative species was confirmed by the restoration of GSH levels in in ethanol fed rats given prickly pear juice.

The successful cellular uptake and internalization of 20 μg/mL of OFI-EVs by HDF cells was confirmed using green dye PKH67 following a fluorescent assay.

All the above experiments validated the potential of OFI-EVs as a natural candidate for healing chronic cutaneous wounds.   

The research idea is a recent hot topic as the use of extracellular vesicles in tissue regeneration is drawing the researchers’ interest. The experimental methods are adequate to some extent. The results are presented and discussed in a good way. The references are up-to-date and the sections are connected in a logical layout. In addition, the conclusion withdrawn are to-the-point.

Recommendation:

I recommend the following modifications to the manuscript:

1.     The authors need to perform an in-vivo histopathological assay using an experimental wound model to validate the biocompatibility and the wound-healing activity of OFI-EVs.

2.     Future perspectives are to be included at the end of the conclusion to find ways for further work to add to this topic in the future.

Comments on the Quality of English Language

Moderate editing of English language required.

Author Response

(The authors gave the same response as above.)

Reviewer 3 Report

Comments and Suggestions for Authors

Dear authors and editor,

The manuscript titled "Extracellular vesicles derived from Opuntia ficus-indica fruit (OFI-EVs) speed up the normal wound healing processes by modulating cellular responses" aimed to demonstrate the ability of Opuntia ficus-indica fruit-derived EVs (OFI-EVs) on the inflammatory process and oxidative stress in an in vitro model of chronic skin wounds, as well as its biocompatibility. 

There are many minor issues I'd like the authors resolve.

Abstract

1-Change the keywords. Delete the words "Opuntia ficus indica" ; "extravesicles"; "antioxidant property" ;"anti-inflammatory activity"; "pro-angiogenic activity" and "chronic cutaneous wounds".  Not found in the MeSH (Medical Subject Headings). It is recommended to include words collected in the thesauri for better dissemination of the manuscript (Opuntia....) Keywords are essential to improve the dissemination of your article within the scientific community.

2-It is recommended to add the study design to the title. 

Introduction

3-Adequate: The most important concepts of the subject to be developed are identified.However, I do not consider it appropriate to end the introduction with the conclusions of the study.

"In this study, the ability of extracellular vesicles isolated from OFI fruits (OFI-EVs) to elicit wound-healing properties was demonstrated for the first time.

Physical properties of OFI-EVs such as concentration and size distribution indicate a highly concentrated particle groups with size range around ∼120 nm. According to our findings, OFI-EVs displayed no effects on cell viability, decreased ROS production and increased proliferation and migration of fibroblasts, while increasing the tube formation capacity of Human Umbilical Vein Endothelial Cells (HUVEC).

To summarize, OFI-EVs are a promising cell-free therapeutic tool for wound healing. However, further studies in in vivo wound healing models should be conducted."

I recommend ending the introduction with the researchers' hypothesis or objectives.

Materials and Methods

4-I recommend expanding the section on statistical analysis.

Results

5-The results are clear and relevant. It is recommended to improve the quality of Figure 2,3,4 and 5 .The graphics are too small to visualise the content.

Discussion

6-The discussion is adequate, the authors comment on and support their postulates. However, I recommend adding a section on limitations.

Conclusion

7-Adequate:The objectives are answered in the conclusions.

Reference

 8-adequate: Complies with the journal's standards.

Congratulations to the authors for their innovative research. This is a topic of interest for improving the healing of chronic wounds. I consider it relevant for future work to apply in vivo, taking into account the aetiology of the injury.

Author Response

(The authors gave the same response as above.)

Reviewer 4 Report

Comments and Suggestions for Authors

Ms. ID: ijms-3039966
Title: Extracellular vesicles derived from Opuntia ficus-indica fruit (OFI-EVs) speed up the normal wound healing processes by modulating cellular responses

Reviewer comments

The target application for these extracellular vesicles (ECVs) is defined at the very end of the abstract using a rather vague language. It should be rewritten for the audience's transparency and awareness, explicitly addressing the chronic and non-healing nature of the wounds. This is the niche for these materials envisioned by the authors, but it is poorly highlighted in the abstract.

It is an interesting manuscript that embarked on an exploration of plant-based ECVs for skin therapy where there are complications caused by inflammatory processes. The authors use a somewhat standard battery of tests to evaluate both inflammatory and oxidative stress markers. Their variations are related to the treatment with ECVs, and the results are encouraging, considering that it is a topic that is rarely explored by other research groups. There are no major issues; this manuscript needs more work on the quality of figures and a more focused narrative.

Technical note to the editorial office staff and authors: line numbering would be extremely helpful during the review.

 Specific comments

Abstract: “OFI-EVs showed high biocompatibility and protective role on the inflammatory process and oxidative stress in an in-vitro model of chronic skin wounds.”  The term biocompatibility is strongly contextual and should be used with the said context in mind. In this case, a non-toxic (or other synonymous term) should be used instead. The toxicity of exogenous materials is of utmost importance for chemical species intended to operate in the intra- and intercellular spaces. What kind of route of administration would you envision for these materials? If it is transdermal administration and the target organ is skin, it should be mentioned in the abstract, to give the reader a preview of what is in the article.

Page 2: “The cutaneous wound healing process is a very sophisticated mechanism that relies on the intricate interaction…” The word mechanism is not needed in this sentence. Two reasons: 1) linguistic, and 2) biomedical: there are likely numerous mechanisms behind wound healing.

Page 2: “The attention is focused on their ability to be natural vehicles of bioactive molecules with high added value for human health [7,8].” This sentence seems out of focus/context and should be rewritten for clarity. What does it mean “on their ability to be natural vehicles”?

Page 2: “P-EVs did not exert cellular toxicity, reducing adverse reactions, particularly when compared to synthetic liposomes and EVs sourced from animals [9].” Every substance is toxic, but it depends on the dosage. Are there any toxicology studies that demonstrated these materials are indeed non-toxic? What about their chronic effects? Is there anything reliable in the peer-reviewed literature, or are scientists just riding the hype of novel therapeutic drugs?

Page 2: “Their therapeutic effects have been demonstrated in different human diseases, including wound healing and tissue repair relying on their powerful antioxidants and anti-inflammatory effect [11].” How does one grade the scale of the antioxidant and anti-inflammatory effects? Are these really “powerful,” as you describe? What does that mean?

Page 3: “…and restoration of GSH levels was also observed [18].” The GSH acronym is not explained.

Page 3: “Ultracentrifugation can yield highly pure preparations by…”. High purity or highly purified.

Page 5: “Figure 1. Isolation and characterization of…”. Please make sure the quality of the figure meets the publisher's standards. This figure is pixelated, and some readers might find it hard to decipher due to the relatively large information density (small images).

Page 6: “Figure 2: Phytochemical analysis…”. The same comment applies here as for Figure 1. The quality is inadequate, especially the illegible chromatogram, which should be replotted using the experimental data. The data in the table should be presented in the descending order of the abundance. You might want to consider a presentation that would reduce the white space in this figure.

Page 8: “Figure 3: In vitro biocompatibility…” Use a larger font for tick and axe labels. These are way too small and difficult to read. Also, I’d suggest exporting the images with at least 300 dpi, if not 600 dpi resolution, and then embedding them in the manuscript. That might help resolve the issue with the pixelated images.

Comments on the Quality of English Language

Noted some minor issues, as illustrated in the "Specific comments".

Author Response

(The authors gave the same response as above.)
